# Parental feeding practices and the relationship with parents in female adolescents and young adults with eating disorders: A case control study

Maria Gruber[1☯], Daniel König[1☯], Julika Holzhäuser[2], Deirdre Maria Castillo[3], Victor Blüml[4], Rebecca Jahn[1], Carmen Leser[3], Sonja Werneck-Rohrer[5☯], Harald Werneck[2☯]*

1 Division of Social Psychiatry, Department of Psychiatry and Psychotherapy, Medical University of Vienna, Vienna, Austria, 2 Department of Developmental and Educational Psychology, Faculty of Psychology, University of Vienna, Vienna, Austria, 3 Department of Obstetrics and Gynaecology, Medical University of Vienna, Vienna, Austria, 4 Department of Psychoanalysis and Psychotherapy, Medical University of Vienna, Vienna, Austria, 5 Department of Child and Adolescent Psychiatry, Medical University of Vienna, Vienna, Austria

☯ These authors contributed equally to this work.
* harald.werneck@univie.ac.at

**Data Availability Statement:** Dataset has been published under https://data.mendeley.com/datasets/cfy4ntg9w3/1.

## Abstract

### Objective

Perceived parental influence on diet in early adolescence in the context of the parental relationship had previously not been studied in a clinical sample. The aim of this study was to investigate a possible association between eating disorders and characteristics of the relationship with parents and the parental feeding practices in early adolescence.

### Methods

21 female adolescents and young adults with an eating disorder (ED)–bulimia nervosa or anorexia nervosa–and 22 females without eating disorder (healthy control; HC), aged between 16 and 26, were assessed via self-report questionnaires for problematic eating behaviour, relationship with parents, perceptions of parent's feeding practices at the age of 10–13 years and personality. Statistical evaluation was performed by means of group comparisons, effect sizes, regression analyses and mediator analyses.

### Results

Adolescent and young adult females with ED reported more fears/overprotection and rejection/neglect by their mothers and less self-responsibility in terms of eating behaviour during adolescence than did the HC. The relationship with the fathers did not differ significantly. Females who perceived more cohesion, rejection/neglect and fears/overprotection by the mother were more likely to suffer from an ED. Rejection/neglect by both parents were associated with less self-acceptance of the young females with even stronger effect sizes for the

**Funding:** Open access funding provided by University of Vienna.

**Competing interests:** The authors have declared that no competing interests exist.

**Abbreviations:** AN, anorexia nervosa; BMI, Body Mass Index; BN, bulimia nervosa; CFQ-A, Child Feeding Questionnaire for adolescents; EBF-KJ, German Version [Elternbildfragebogen für Kinder- und Jugendliche]; ED, eating disorder; EDI-2, Eating Disorder Inventory-2; HC, healthy controls; ICD-10, International Classification of Mental and Behavioral Disorders 10th Edition; PRSQ, Parental-Representation-Screening-Questionnaire.

fathers than the mothers. Harm prevention in the young females was a partial mediator between fears/overprotection and the drive for thinness.

## Conclusions

The parental relationship is partly reflected in the self-acceptance and self-responsibility in eating of the adolescent and young females, both of them are particularly affected in EDs. Stressors in the parent-child relationship should be targeted in treatment of eating disorders. Nutritional counselling for parents might be useful in early adolescence.

## Introduction

Feeding and eating disorders (ED) are characterized by disturbances of eating and other behaviour, leading to changes in food consumption and impairing physical health or psycho-social functioning: Anorexia nervosa (AN) is defined by restrictions in energy intake in relation to need, resulting in significantly low weight. Bulimia nervosa (BN) describes recurring episodes of binge eating and inappropriate compensatory behaviour to prevent weight gain. In both disorders, self-evaluation is affected and strongly influenced by the way one experiences one's body shape and weight [1].

ED mainly affect females and exhibit a significant increase of incidence during adolescence [2]. Lifetime prevalence of AN and BN for females is reported to be 2.9% (0.1% among males of the same age brackets) [3]. A longitudinal cohort study on children, adolescents and young adults in Germany [4] found that 21.9% of the 11 to 17 years old even showed symptoms of an ED. Furthermore, the rates of incidence as well as prevalence of AN among adolescents have been increasing significantly in recent years [5] particularly in the highly vulnerable period before or at the beginning of adolescence [6]. Pubertal development, with its physical changes, often precedes or accompanies adolescence and early maturing girls are reported to be at special risk from disordered eating [7]. Adolescence involves several major developmental tasks such as adapting to physical changes, creating one's own identity, defining goals for the future or learning self-regulation of emotions which present stressful situations for young girls [2].

The bio-psycho-social model for the development of psychiatric disorders [8], representing the theoretical background of this study, postulates, that the genetic vulnerability for developing a certain psychiatric disorder, together with the intrapersonal stress and the environmental stress factors (family, friends, schooling situation), may lead to the outbreak of a disease.

Since adolescence represents a period of the life cycle characterised by progressive independence from parents and family members, it may be particularly valuable to examine the extent to which family factors influence the behavioural and emotional wellbeing of the adolescents and consequently to the risk of developing an ED [2]. Subjectively perceived stressors in the parental relationship are hypothesized to increase the risk for psychosocial maladaptation and the occurrence of EDs [9]. Furthermore, the development of abilities to regulate emotions and impulses is influenced by the parental relationship [10] and the capacity to self-regulate was shown to be negatively affected by stronger parental control [11].

Attachment theory provides a comprehensive framework for understanding the psychological factors and family characteristics contributing to individual development in adolescence [12]. Early interactions and experiences with parents are internalized and used to regulate self-esteem. According to the attachment theory, secure attachment with availability, understanding and responsiveness of caregivers leads to a healthy self-esteem, while negative or inconsistent parenting behaviour is more likely to lead to problems in self-assessment [12].

The relationship between children and their parents instinctively shows different forms of bidirectional communications in different contexts and aims. Especially the feeding situation seems to play an important role in child-parent attachment. It has been shown that during feeding, when the relationship between children and parents is not able to coordinate and share sensitive interactions, the infant will not be able to learn the self-regulation of affects and may show maladaptive emotional and behavioural symptoms over time [13]. These parenting practices and their influence on children's emotional as well as food self-regulation abilities [14] are believed to play an important role in the development and maintenance of EDs [10, 14].

Parental overprotection during childhood is thought to lead to increased awareness of stress and mental health problems in adulthood [15]. Both, fathers and mothers of persons with EDs, are reported to overprotect their children [16]. However, while females with AN more frequently than their healthy siblings reported to have experienced maternal overprotection in their childhood, no such difference was reported for females with BN [17]. Contradictory, parents of patients suffering of AN [18–20] or BN [19] are postulated to be perceived as less caring, highly disengaged, poorly interwoven, rigid and with less cohesion and communication qualities, in comparison to parents of the respective peer group. It was also previously demonstrated, that persons with EDs experienced strong entanglement within their families [21]. To better understand parental impact on adolescents' eating disorders, it is necessary to address the influence of parental feeding practices on eating behaviour. The current study addresses the limitations in the existing literature in several ways: Female patients with eating disorders (ED) and healthy controls (HC) were asked for both, their recent relationship with their parents and for their memories of parental feeding practices at early adolescence, retrospectively.

The first aim was to examine female adolescent and young adults with ED as well as the HC group for their contemporary perception of the relationships with their mother or father. Dysfunctional relationships with parents in females with ED are reported in literature, but previous data are contradictory. We expected the adolescent and young females with ED to report a significantly worse relationship with their mother or father than the HC.

The second aim was to explore a possible mediating role of the personality traits ("self-directedness", "self-acceptance" and "harm avoidance") on the maternal relationship and symptoms of eating disorders ("drive for thinness" and "body dissatisfaction"). Recent findings suggest self-regulating skills, such as self-compassion, reduce the effects of perfectionism on the development of ED [22]. Therefore, in the present study we anticipated a possible positive influence of personality traits on maternal relationship and eating disorder symptoms.

The third aim was to retrospectively evaluate the perceived parental influence on eating behaviour at the age of 10–13 years. Previous studies investigating the connection between the influence of parents' feeding practices and anorexic or bulimic EDs have primarily focused on early childhood and are still under-represented in research [23]. To the best of our knowledge, no previous study exists investigating the perceived parental' feeding practices in early adolescence in the context of the parental relationship in a clinical sample of females diagnosed with ED. However, controlling food-related parenting practices were associated with adolescent disordered eating behaviours in population-based samples [24], and restrictive parental feeding practices were shown to lead to disordered eating behaviour in children in longitudinal studies [25, 26]. We expected remembered parental feeding practices in terms of "feeding responsibility", "monitoring", "pressure to eat" and "restriction" to be significantly more frequent in females with EDs than in HC.

## Methods

### Subjects and procedures

**Study design.** This quantitative cross-sectional study aimed at examining the current situation and memories of the early adolescence in female adolescents and young adults with and without eating disorders (ED) using questionnaires. The data was collected in the period from September 2016 to February 2017. Although no longitudinally collected data was available, cause-effect conclusions could be drawn by using a structural modelling strategy [27].

**Ethics and consent to participate.** This study was approved by the ethics committee of the Medical University of Vienna (reference number: **1691/2016**). Written consent by the participants, after thorough information about the study, was acquired in advance in all cases. Adolescents between 14 and 18 years of age are of legal age of consent in Austria. Thus, young females' consent in our study population of older than 16 years, was sufficient–and in accordance with the Ethics Committee–and parents did not have to be informed about participation in the study.

**Study setting.** Female adolescents and young adults with ED were recruited from inpatient and outpatient care of the Clinical Division of Social Psychiatry at the Department for Psychiatry and Psychotherapy, and of the Department for Child and Adolescent Psychiatry of the Medical University of Vienna. The healthy control group (HC) was recruited via circle of acquaintances.

**Study sample.** *Female adolescents and young adults with eating disorder.* Females with an eating disorder (ED) were included if they a) were between 16 and 26 years of age b) were diagnosed with either the ICD-10 diagnosis bulimia nervosa (BN) (ICD-10: F50.2) or anorexia nervosa (AN) (ICD-10: F50.00)—restrictive type or AN—bulimic type (ICD-10: F50.01). The diagnoses were established by experienced clinicians at the recruitment institutions.

*Healthy controls.* Inclusion criteria for the healthy control group (HC) were a) the willingness to participate and b) sufficient German language skills. Exclusion criteria for the HC were a) the existence of a current or past diagnosis of an ED, b) values above the normal range in at least two of the three scales: "drive for thinness", "bulimia", "body dissatisfaction" of the Eating Disorder Inventory-2 (EDI-2) [28]. When recruiting the control group, a similar age distribution as in the group of females with eating disorders was aimed at, but no age matching was performed in the analysis.

### Measurements

**Demographic data** of the participants were collected by means of a study specific questionnaire. Data about age, weight, diagnosed mental health disorders according to ICD-10 [29], participation and duration of psychological/psychiatric treatment, housing situation with/without parents and school degree were collected. The symptoms of EDs were investigated using the frequently used **Eating Disorder Inventory 2 (EDI-2; German Version)** [28]. The EDI-2 is a revision of the EDI-1 [30], in which three subscales have been added. In the original English version of the EDI-2, the internal consistencies (Cronbach's Alphas) for the eight original scales ranged from 0.80 to 0.91, of which three were used in this study and a Principal Component Analysis supported the factor structure of the original EDI, but not that of the three additional scales [31] not used in this study. For our study, the validated German version of the questionnaire was used, which contains 91 items on a 6-level rating scale representing 11 scales. The German version achieved an Cronbach's Alpha between $\alpha = 0.73$ and $\alpha = 0.93$ in a sample of anorexic and bulimic patients, the test-retest reliability ranged from rtt = 0.81 to rtt = 0.89 [28]. In the current study, the three subscales (23 items) with the most relevant

content for the symptoms of eating disorders were applied and Cronbach's Alpha of these scales reached $\alpha = 0.91$ for "drive for thinness", $\alpha = 0.95$ for "bulimia" and $\alpha = 0.93$ for "body dissatisfaction" in the study population. The quality of the relationship with the parents was measured with the **Parental-Representation-Screening-Questionnaire (PRSQ); German Version [Elternbildfragebogen für Kinder- und Jugendliche (EBF-KJ)]** for children and adolescents [32]. This German questionnaire, which was applied in this study in its original version, consists of 36 items answered on a 5-level Likert scale and contains three resource-scales ("cohesion", "identification", "autonomy"), five risk-scales ("conflicts", "rejection/neglect", "punishment", "emotional burden", "fears/overprotection") and one additional scale "aid". The Cronbach's Alpha of this validated version were between $\alpha = 0.60$ and $\alpha = 0.85$ for students and between $\alpha = 0.73$ and $\alpha = 0.90$ for patients [32]. Due to the quantitative evaluation of our study, items with an open response form were not used and for reasons of content, three subscales were not included for analysis of this study (25 instead of 36 items were evaluated). In the current sample, following reliabilities were measured related to the mother or the father: "cohesion" $\alpha = 0.92$ or $\alpha = 0.89$, "identification" $\alpha = 0.80$ or $\alpha = 0.87$, "autonomy" $\alpha = 0.69$ or $\alpha = 0.81$, "conflicts" $\alpha = 0.85$ or $\alpha = 0.83$, "punishment" $\alpha = 0.82$ or $\alpha = 0.54$, "rejection/neglect" $\alpha = 0.90$ or $\alpha = 0.81$, "emotional burden" $\alpha = 0.77$ or $\alpha = 0.84$, "fears/overprotection" $\alpha = 0.84$ or $\alpha = 0.85$ and "aid" $\alpha = 0.73$ or $\alpha = 0.56$.

The **Child Feeding Questionnaire for adolescents (CFQ-A)** [33] which is an adaption from the CFQ for parents of adolescents [34] and the CFQ for parents of children ranging in age from about 2 to 11 years [35], was used to assess the perceptions of the feeding practices that parents/caregivers used when participants were between 10 and 13 years of age. The Cronbach's Alpha of the English original version of the CFQ-A were $\alpha = 0.68$ for the scale "responsibility", $\alpha = 0.90$ for "monitoring", $\alpha = 0.63$ for "pressure to eat" and $\alpha = 0.85$ for "restriction" [33]. For the purpose of this study, the CFQ-A questionnaire was translated into German language. The 19 items rated on a Likert scale are reflected on four subscales with following internal consistencies in the present sample: "perceived feeding responsibility" (after elimination of item 1) $\alpha = 0.80$, "monitoring" $\alpha = 0.90$, "pressure to eat" $\alpha = 0.84$ and "restriction" $\alpha = 0.87$.

**The Junior Temperament and Character Inventory 12–18 R** [36] provides an evaluation of personality in adolescents. The questionnaire consists of 103 items, which are answered on a Likert scale. The German original version of the questionnaire showed internal consistencies between $\alpha = 0.79$ and $\alpha = 0.85$ [36]. Only several contently relevant scales were evaluated. In the current study an internal consistency of $\alpha = 0.90$ for "harm avoidance", $\alpha = 0.91$ for "self-directedness" and $\alpha = 0.90$ for "self-acceptance" was calculated.

## Statistical analysis

The data was evaluated using IBM SPSS Statistics 24. The parental relationship of females with ED and HC were calculated using the t-test for heterogeneous variances (Welch test) and effect sizes according to Cohen. To examine the association of an ED with the relationship with the parent, a binary logistic regression with a stepwise backward method according to Wald was calculated in the total sample (ED and HC). Nagelkerke's $R^2$ was used to check the classification quality. A value > 40% can be regarded as good. The relationship between "rejection/neglect" and "self-acceptance" was calculated using the Spearman correlation (two-sided testing). The mediator analysis was calculated in the total sample (ED and HC) and therefore PROCESS 2.16.3 according to Hayes for SPSS® was applied with the criteria "drive for thinness" in relation to maternal relationship quality.

The parental influence on diet at the age of 10–13 years for females with ED and HC was compared bilaterally using the Welch test.

A power calculation was performed on the basis of the assumption of a difference of 4.5 points between females with ED and HC on the subscale "rejection/neglect" of the Parental-Representation-Screening-Questionnaire (PRSQ) and a confidence level of 95%. Mean values of school kids with normal parental representation (M 49 ± 4.8) and a psychiatric patient group with normal and abnormal parental representations (M 53.5) were derived from literature [9]. This showed that to obtain a power of 80% for the detection of significant differences, a minimum sample size of 18 patients per group was required.

## Results

### Sample description

A total of 52 females were included in the study. Of these, 30 participants were included into the healthy control group (HC), while 22 adolescent and young adults were included into the eating disorders group (ED). In the group of participants in whom clinicians diagnosed an eating disorder (ED) according to the criteria of the ICD-10, one participant had to be excluded due to lack of data and 21 persons could be subjected to further analyses. In the healthy control (HC) group, 8 participants were excluded as they were classified as "at risk of developing an ED" due to symptoms of disordered eating (according to EDI-2). The data of 22 HCs could be further analysed. Due to missing values, the sample size varied in the individual calculations.

The age of the study participants was between 16 and 26 years with a similar distribution in both groups (HC: M = 18.55 ± 2.67; ED: M = 18.86 ± 2.72). The Body Mass Index (BMI) was M = 22.63 for the HC group and M = 18.30 for the ED group. A BMI less than 18.50 was classified as underweight.

In the ED group, 17 (80.9%) females had been diagnosed with anorexia nervosa while four (19.1%) females had been diagnosed with bulimia nervosa. One person from the ED-group also suffered from depression. The onset of the ED was between 10 and 19 years (M = 15 ± 1.97). Of the participants with an ED, 21 (100%) were currently in psychological or psychiatric treatment. Persons in the HC group did not suffer from any diagnosed mental health disorder.

**Relationship with parents.** With regard to the quality of the relationship with the mother, significant differences were revealed between the groups (ED n = 21, HC n = 22) for the scales "rejection/neglect" (t(24.75) = - 2.13, p = 0.043, d = - 0.65) and "fears/overprotection" (t(38.72) = - 3.01, p = 0.005, d = - 0.92). The results are demonstrated in Table 1. No significant differences could be demonstrated for the relationship with the mother for each group in "cohesion" (ED and HC) (t(40.32) = 0.95, p = 0.394, d = 0) 0.29), "autonomy", (t(40.96) = 0.67, p = 0.504, d = 0.20), "conflicts" (t(40.85) = - 0.83, p = 0.410, d = - 0.25), and "emotional burden" (t(39.94) = 0.37, p = 0.714, d = 0.11). There was no significant difference in any scale between these two groups in terms of the quality of the relationship with the father.

**Association of an eating disorder: Relationship with the mother.** The variables "conflicts", "emotional burden", "autonomy", "cohesion", "rejection/neglect" and "fears/overprotection" of the mother were investigated by means of binary logistic regression (n = 42; see Table 2). It was found that young females who perceived more "cohesion", "rejection/neglect" and "fears/overprotection" by their mother more often suffered from an ED. The model explained 43.9% of the variance.

**Mediators between maternal relationship and symptoms of eating disorders.** The mediation model tries to identify a possible process that underlies the observed relationship between the independent variable "relationship with the mother" and the dependent variable "symptoms of eating disorders" via the inclusion of a third hypothetical mediator variable "personality traits". In this study, analyses were only carried out for variables with strong

**Table 1. Differences in relationship quality with the mother between females without eating disorders (HC) and females with eating disorders (ED).**

| Scale | Mean value (SD) | | t | p | d |
|---|---|---|---|---|---|
| | HC | ED | | | |
| | (n = 22) | (n = 21) | | | |
| Cohesion | 3.12 (±0.96) | 2.83 (±1.04) | 0.95 | 0.349 | 0.29 |
| Autonomy | 2.89 (±0.69) | 2.75 (±0.64) | 0.67 | 0.252 | 0.20 |
| Conflicts | 1.77 (±0.85) | 1.97 (±0.76) | - 0.83 | 0.410 | -0.25 |
| Rejection/neglect | 0.11 (±0.25) | 0.46 (±0.71) | - 2.13 | 0.022* | -0.65 |
| Emotional burden | 1.35 (±0.75) | 1.26 (±0.85) | 0.37 | 0.714 | 0.11 |
| Fears/overprotection | 1.58 (±0.78) | 2.38 (±0.95) | - 3.01 | 0.003* | - 0.92 |

Average values, standard deviations (±SD), *p-values* and effect strengths (*d*) of the relationship to the mother are displayed.

Based on values from 0 to 4, meaning very low to very high.

*p < 0.05, one-sided.

correlations in previous calculations. For eating disorder symptoms, a separate analysis was done with the criteria "body dissatisfaction" and "drive for thinness" since it was not reasonable to combine these two scales to a common scale. Since there was no significant association between "autonomy" and the mediator "self-directedness" (r = 0.08, p = 0.627, n = 42), nor between "autonomy" and the criteria "body dissatisfaction" (r = - 0.08, p = 0.586, n = 43) / "drive for thinness" (r = 0.00, p = 0.995, n = 43), there can be no mediation and therefore, no analysis was carried out. The applied scales were tested for z-standardization before the analysis and with regard to the procedural requirements, the normal distribution of the data could generally be assumed on the basis of the validity of the central limit theorem for sample sizes $\geq$ 30.

The mediator analysis (n = 42) found a direct and an indirect effect of "fears/overprotection" on the "drive for thinness" via anxious personality traits ("harm avoidance"). The direct effect measures the extent to which "fears/overprotection" changes when "drive for thinness" increases and the mediator variable remains unaltered. In contrast, the indirect effect measures the extent to which "drive for thinness" changes when "fears/overprotection" is held fixed and "harm avoidance" changes by the amount it would have changed had "fears/overprotection" increased by one unit. Part of the effect was mediated indirectly through the mediator "harm avoidance" (β = 0.14) and part of the effect was directly mediated from "fears/overprotection" to "drive for thinness" (β = 0.41), together they had a medium effect (β = 0.55). Fig 1 shows the relationships embedded in the model and gives the statistical parameters of the effects. The effect of "fears/overprotection" on "body dissatisfaction" was not mediated by "harm avoidance" (β = 0.07).

**Table 2. Predictors for the presence of an eating disorder.**

| | B | SE | Forest χ² (df = 1) | p | OR |
|---|---|---|---|---|---|
| Cohesion | 1.40 | 0.72 | 3.73 | 0.053 | 4.04 |
| Rejection/Neglect | 3.38 | 1.49 | 5.15 | 0.023* | 29.35 |
| Fears/Overprotection | 1.45 | 0.54 | 7.07 | 0.008* | 4.24 |
| Constant | - 7.92 | 3.18 | 6.21 | 0.013* | 0.00 |

Coefficients and test quantities of binary logistic regression (using the exclusion method).

Significance level: α = 10%. Excluded: Conflicts, "emotional burden", autonomy. *B* = unstandardized regression coefficient. *OR* = Odds ratio. *n* = 43.

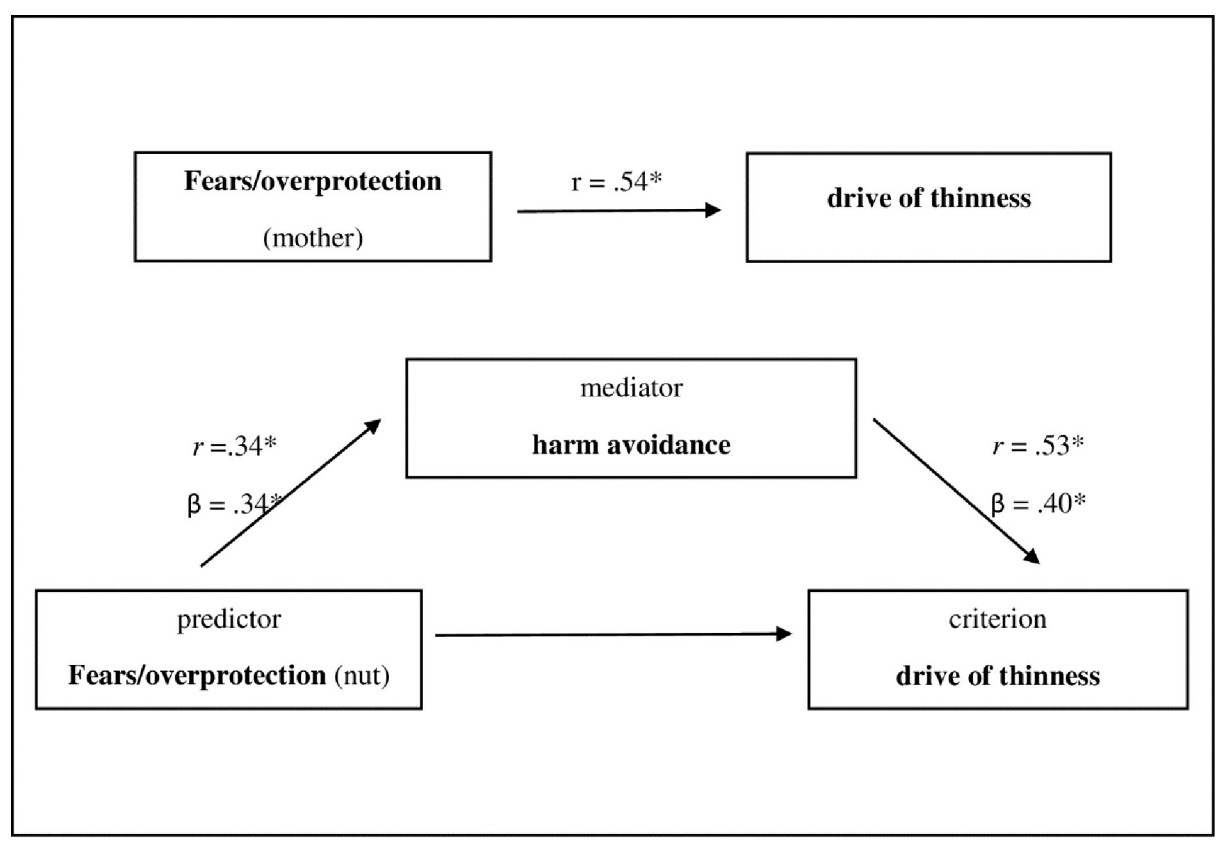

**Fig 1. Mediator model fears/overprotection—harm avoidance—drive for thinness.** *p<0.05. n = 42. β = standardized regression coefficient, indirect effect β .14, boot SE 0.08, BootLLCI (t) 0.01 a, BootULCI (p) 0.36 a, ªconfidence interval 0 not included, therefore effect significant Bootstrap: 5000.

**Parental feeding practices at the age of 10–13 years.**   The scales "monitoring" (t(39.41) = - 0.78, p = 0.443, d = - 0.24), "pressure to eat"(t(33.03) = - 1.55, p = 0.131, d = - 0.47) and "restriction" (t(35.35) = - 1.55, p = 0.129, d = - 0.47) showed no significant differences. Only the scale "perceived feeding responsibility" showed a significant difference between the two groups (t(39.67) = - 2.20, p = 0.034, d = - 0.67). Female adolescents and young adults with an ED reported that they had less personal responsibility at the age of 10–13 years for what and how much they ate in comparison to females without an ED (see Table 3).

**Table 3. Comparison of perceived parental influence on diet at the age of 10–13 for females without eating disorder (HC) and females with eating disorder (ED).**

| Scale | Mean value (SD) | | t | p | D |
|---|---|---|---|---|---|
| | HC | ED | | | |
| | (n = 22) | (n = 21) | | | |
| Perceived feeding responsibility | 2.20 (0.93) | 2.88 (1.07) | - 2.20 | **0.034***  | - 0.67 |
| Monitoring | 2.95 (1.10) | 3.24 (1.29) | - 0.78 | 0.443 | - 0.24 |
| Pressure to eat | 2.37 (0.77) | 2.87 (1.24) | - 1.55 | 0.131 | - 0.47 |
| Restriction | 1.96 (0.68) | 2.36 (0.98) | - 1.55 | 0.129 | - 0.47 |

Based on values from 1 to 5, meaning very low to very high. p ≤ 0.05, two-sided.

**Relationship between parental rejection/neglect and self-acceptance.** A significantly negative correlation between the scale "rejection/neglect" of the mother and "self-acceptance" of the test subjects could be determined (rs = - 0.35, p = 0.022, n = 42). There was also a significant negative correlation between the fathers' "rejection/neglect" and "self-acceptance" (rs = - 0.47, p = 0.002, n = 42). This means that more rejection/neglect by both the mother and the father was accompanied by less self-acceptance of young females with an even greater influence of the fathers' rejection/neglect.

## Discussion

The aim of the study was to investigate a possible association between eating disorders (ED) and subjective stress in the relationship with parents as well as retrospective parental influence on eating behaviour at the early adolescence from the perspective of affected young females.

Adolescents and young adult females with ED reported more fears/overprotection and rejection/neglect by their mothers and less self-responsibility regarding eating behaviour in adolescence than healthy young females. The relationship with fathers did not differ significantly. It was shown that young women who perceived more cohesion, rejection/neglect and fears/overprotection from their mothers had an increased chance of suffering from an eating disorder. Rejection or neglect by the mothers and the fathers was associated with lower self-acceptance among young females, while the influence of father's rejection/neglect on self-confidence was even greater.

According to the attachment theory, a well-functioning relationship with parents is crucial for children to learn self-acceptance and independent emotion regulation in adolescence–both of which are known to influence the development of eating disorders in adolescence [13]. Previous research have focused on parental functioning in families of patients with ED, with a particular emphasis on the links between the psychopathological symptoms of mothers and daughters. However, only few and often contradictory findings exist on whether there are specific maladaptive psychological profiles characterizing the family structure as a whole when it includes adolescent and young adult females with anorexia nervosa (AN) or bulimia nervosa (BN) [37]. In our study, we expected to find an association of EDs and problematic family functioning. Furthermore, we expected females with an ED to have experienced significantly more rejection/neglect or fears/overprotection from their mothers than females without an ED.

While rejection/neglect or fears/overprotection are contrary relationship patterns, our results indicate that mothers of patients with ED are experienced in different extremes between entanglement and rejection–in part depending on the type of ED.

In previous literature, adolescents with BN were reported to having had a significantly more dysfunctional family background than those with AN [11]. Patients with BN perceived their families as badly organized, and reported the presence of high levels of family conflicts [37]. These conflicts may have led to perceived rejection/neglect from parents. In contrast, in families of adolescents diagnosed with AN, higher conflict avoidance and enmeshment was found. This means that over-concern for others with a loss of autonomous development could lead to fears/overprotection [37]. Literature, however, remains inconclusive as both a tendency towards parental overprotection in EDs [38] as well as no connection between overprotection and EDs have previously been demonstrated [39]. Our results are in agreement with previous literature where a general tendency towards more perceived parental rejection in persons with an ED was postulated [16, 18, 19].

The results regarding the father-daughter relationship and eating disorders are of particular interest. A recent review [40] identified a gap in the literature regarding this topic and called

for research to further investigate this dyadic relationship. The father-daughter relationship is important during adolescence as it plays a role in the development of autonomy and self-esteem, both of them are involved in eating disorder psychopathology [40].

In our study, it could be shown that rejection or neglect by both mother and father was accompanied by less self-acceptance on the part of the young females. No direct connection could be found between the quality of the relationship with the father and the presence of an ED. However, an indirect effect of fathers on the symptoms of EDs was shown, since subjectively perceived rejection or neglect by the father was accompanied by less self-acceptance on the part of the young females. The paternal influence on self-acceptance was even greater than that of the mothers. Interestingly, a recent study found that changes in the perceived quality of the father–adolescent daughter attachment relationship were only associated with daughters' self-esteem (not to that of the sons') [41]. As attachment theory indicates, exploration and security represent two sides of a secure attachment style [12], and, according to traditional role models, fathers are seen as more independence encouraging (while mothers serve as a haven of safety). But some changes to the father–daughter attachment can be observed when girls approach adolescence. In adolescence, daughters may increasingly start to value their fathers' independence-encouraging behaviour. This makes them more susceptible for shortcomings in the perceived quality of the father–child attachment relationship [12, 40].

Harm avoidance as an adolescent and young adult females' personality trait in part mediated the direct influence of the fears/overprotection of the mothers on aspects of the ED symptoms of the young females. Parents exercised some control over the young females' drive for thinness through experienced overprotection, which was influenced by their own control in the form of harm avoidance. Stenbæk [15] found a similar influence of the personality trait harm avoidance, which had a mediating effect on the direct influence of parental overprotection and perceived stress or susceptibility to psychological symptoms in healthy adults. It has also been shown that self-regulating skills, such as self-compassion, moderate the impact of perfectionism on disordered eating [22].

In this study, in advance, we expected that female adolescents and young adults with ED will report about more experienced and stricter regulations on dieting than HC. Interestingly, the female adolescent and young adults with EDs reported significantly less feeding responsibility in early adolescence compared to the females in the HC group, but other characteristics of controlling parental feeding practices like food restrictions, monitoring during meal and the pressure to eat did not show significant difference in both groups. Previous studies had shown restrictive parental feeding practices to increase the likelihood of disordered eating [25, 26]. If excessive parental control over food consumption is applied in raising children, the development of self-regulatory capacities could be negatively influenced [25]. Stringent maternal restriction of food resulted in eating psychopathology such as extreme weight control in daughters [24]. Although we found no such effect of controlling feeding practices such as restriction, pressure to eat and monitoring in our study, we did find an effect on feeding responsibility, which is the opposite of controlling practices of the parents by informing about the freedom to make dietary decisions on behalf of the adolescent females.

## Strengths and limitations

A strength of our paper is the innovative study design investigating the perceived parental' feeding practices in early adolescence in the context of the relationship with the parents in a clinical sample of female adolescent and young adults diagnosed with EDs as well as in healthy participants (HC). To the best of our knowledge, no examination of perceived parental feeding practices, as evaluated in this study adapting the Child Feeding Questionnaire for Adolescents

[35], in females with valid diagnoses of EDs according to ICD-10 [29] has yet been conducted by clinicians at ED institutions. Previous studies were carried out in non-clinical population and they mainly investigated the relationship between parental feeding practices and eating disorders psychopathology [35, 42] The DSM-5 [1] has recently also included binge eating disorder (BED) as an autonomous diagnosis characterized by recurrent episodes of eating fits accompanied by a feeling of loss of control but lower interest in body shape or weight. In the present study, we state that the parental relationship is mediated through self-assessment and that self-evaluation is extremely influenced by the symptoms of the eating disorder (such as body shape or weight). Compared to AN and BN, body shape and weight do not seem to play such a central role in regulating self-confidence in the BED, so the BED was not included.

Some limitations do, however, need to be addressed: The cross-sectional design of the present study does not allow causal interpretation and results need to be interpreted with causation. Although the healthy control (HC) group was composed with regard to a similar age distribution as in the group of female adolescents and young adults with eating disorders (ED), no age matching was performed in the analysis.

Furthermore, although a sample size calculation based on group differences of main characteristics had been performed, a relatively small sample size may be interpreted another limitation of our study as only moderate and clear effects, but no small effects may be recorded [43] and results with very high standard deviations should be treated with caution. However, the significant and reliable differences and effect sizes found despite these limitations underline the relevance of the results. Eight people were excluded from the control group due to symptoms of an eating disorder above the threshold, which means that almost one third were affected by the disordered eating behaviour. Therefore, the results of females with ED were not compared with the baseline values of a group representative of the population. The retrospective collection of the data on perceived parental feeding behaviour in early adolescence of females offers limitations due to possible distortions of the remembered influence [44]. The retrospective measurement is another possible weakness of the study, as we cannot rule out possible recall biases involved in the reports of the adolescent and young adult females aged between 18–19 years making statements about parental feeding practices when they were 10–13 years of age. Another limitation is that the Child Feeding Questionnaire for adolescents (CFQ-A) was only translated into German language for this study. To our knowledge there exists no German validated version and therefore we have provided the psychometric properties of our translated version. While the questionnaire was first translated into German by multiple authors and then back translated into an English version for comparability, the translation was not previously validated.

In this study, based on these observations, we assume that female adolescents and young adults with ED have experienced more or stricter regulations on dieting than HC. As our study does not include longitudinal data, only retrospectively assessed recall may be interpreted in our cross-sectional questionnaire study. Previous literature, however, has shown, that cross-sectional studies may infer information of possible influencing factors[27, 45] Nonetheless, results need to be interpreted with caution.

## Conclusions

As an eating problem manifests and develops, the signs of disordered eating are enormously stressful for the whole family and it is natural for parents to react with grave concerns. Feeding practices and relationships might be drastically altered in profound ways. A recent meta-analysis underlines difficulties in establishing a therapeutic alliance and highlights especially the risk that professionals, adolescents, and parents will not converse about treatment, although such a

dialogue appears to be an essential component in the construction of a therapeutic alliance [46]. As the results of the current study also underline, prevention and treatment efforts with focus on nutritional counselling of parents and stressors in the parent-child relationship should be targeted in particular at young females aged 10–19, at the highest incidence of anorexia and bulimia [44]. The established family therapy of adolescents with EDs [42], which works on dysfunctional parental relationship, and the psychotherapy of adults with EDs, which focuses on self-regulatory mechanism [10], therapeutic relationship and personality development, would also prove useful against the background of the present results. Knowledge about parental influence on eating behaviour and psychosocial development in the context of EDs may be useful in parent counselling and prevention work so that parents can be increasingly seen as a resource in the prevention and treatment of EDs. There is a need for further research with a prospective design and more objective measures like observation by investigators to allow for statements about parental feeding practices/relationship as protective or risk factors in the development of eating disorders.

## Author Contributions

**Conceptualization:** Maria Gruber, Daniel König, Julika Holzhäuser, Deirdre Maria Castillo, Sonja Werneck-Rohrer, Harald Werneck.

**Data curation:** Maria Gruber, Daniel König, Julika Holzhäuser, Sonja Werneck-Rohrer, Harald Werneck.

**Formal analysis:** Daniel König, Julika Holzhäuser, Harald Werneck.

**Investigation:** Maria Gruber, Daniel König, Julika Holzhäuser, Deirdre Maria Castillo, Victor Blüml, Rebecca Jahn, Carmen Leser, Sonja Werneck-Rohrer.

**Methodology:** Daniel König, Julika Holzhäuser, Victor Blüml, Carmen Leser, Sonja Werneck-Rohrer, Harald Werneck.

**Project administration:** Daniel König, Sonja Werneck-Rohrer, Harald Werneck.

**Resources:** Daniel König, Sonja Werneck-Rohrer, Harald Werneck.

**Software:** Daniel König.

**Supervision:** Sonja Werneck-Rohrer, Harald Werneck.

**Validation:** Sonja Werneck-Rohrer, Harald Werneck.

**Writing – original draft:** Maria Gruber, Daniel König, Julika Holzhäuser, Deirdre Maria Castillo, Sonja Werneck-Rohrer, Harald Werneck.

**Writing – review & editing:** Maria Gruber, Daniel König, Victor Blüml, Rebecca Jahn, Carmen Leser, Sonja Werneck-Rohrer, Harald Werneck.

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
