## [Decision Letter · Decision Letter 0]

5 Jun 2020

PONE-D-20-11413

Parental feeding practices and the relationship with parents in young women with eating disorders: A case control study

PLOS ONE

Dear Dr. Werneck,

Thank you for submitting your manuscript to PLOS ONE. After careful consideration, we feel that it has merit but does not fully meet PLOS ONE’s publication criteria as it currently stands. Therefore, we invite you to submit a revised version of the manuscript that addresses the points raised during the review process.

We look forward to receiving your revised manuscript.

Kind regards,

Silvia Cimino

Academic Editor

PLOS ONE

Journal Requirements:

2. Please state in your methods section whether you obtained consent from parents or guardians of the minors (those aged <18) included in the study or whether the research ethics committee or IRB approved the lack of parent or guardian consent.

5. Your ethics statement must appear in the Methods section of your manuscript. If your ethics statement is written in any section besides the Methods, please move it to the Methods section and delete it from any other section. Please also ensure that your ethics statement is included in your manuscript, as the ethics section of your online submission will not be published alongside your manuscript.

Reviewers' comments:

Reviewer's Responses to Questions

**Comments to the Author**

1. Is the manuscript technically sound, and do the data support the conclusions?

Reviewer #1: Partly

Reviewer #2: No

2. Has the statistical analysis been performed appropriately and rigorously? 

Reviewer #1: Yes

Reviewer #2: Yes

3. Have the authors made all data underlying the findings in their manuscript fully available?

Reviewer #1: Yes

Reviewer #2: Yes

4. Is the manuscript presented in an intelligible fashion and written in standard English?

Reviewer #1: No

Reviewer #2: Yes

5. Review Comments to the Author

Reviewer #1: Thank you for the opportunity of revising this manuscript titled “Parental feeding practices and the relationship with parents in young women with eating disorders: A case control study’. The study took in consideration, in a sample of young women with and without an eating disorder, the possible association between eating disorders and characteristics of the relationship with parents and the parental feeding practices in early adolescence.

I read the article and I think the focus is very interesting. Despite this, several structural revisions are necessary. So I think that it can be published in this Journal, but with major revision. In particular I think it is important to give evidence to the theoretical model, also illustrating a major study of recent literature. Please find below some comments.

INTRODUCTION

In the introduction, the concept of EDS is introduced, but it is not adequately explained, referring also to the most recent diagnostic classifications (e.g. DSM-5).

Moreover, the adolescent population is the object of interest. However, the authors reported prevalence rates without referring to this specific developmental stage. It's important to report the prevalence rates and the clinical relevance of the phenomenon in adolescence. Moreover, I suggest to cite previous studies that have shown that the peaks of EDS onset are in the adolescent phase. See, for example, the study by Poppe I, Simons A, Glazemakers I, Van West D. Early-onset eating disorders:a review of the literature. Tijdschr Psychiatr. 2015;57(11):805–814. In addition, in the introduction is not specified why adolescence represents a phase of risk for the onset and maintenance of Eating disorders. It would be important to underline the peculiarities of this specific evolutionary stage. We recommend, for example: Baker et al., (2012), “Pubertal development predicts eating behaviors in adolescence” Or Ballarotto et al.(2017). Does alexithymia have a mediating effect between impulsivity and emotional-behavioural functioning in adolescents with binge eating disorder?. Clinical Neuropsychiatry, 14(4).

Moreover, from the outset, the theoretical framework should be clear. The authors should describe from the beginning the theoretical framework from which they start for their own study.

On the other hand, the links between parents and children are highlighted, but the influence and emotional exchange between parents and children, linked to eating patterns, is not clear. It can be helpful for a review, see the study: Cimino et al.(2018). Impact of parental binge eating disorder: exploring children's emotional/behavioral problems and the quality of parent–child feeding interactions. Infant mental health journal, 39(5), 552-568.

The aims of the study are not clear. For example, the statistical analysis section showed that you also tested a mediation model, but it is not clear on the basis of which hypotheses. I suggest that you reorganize the last part of the introduction, describing the objectives of the study in detail. For each aim, it is also important to specify what the hypotheses are, and based on which previous studies and theoretical perspective.

This is not a longitudinal study, so authors should clarify already in the introduction because, and on the basis of which literature, it is possible to draw cause-effect conclusions in retrospective or cross-sectional studies. See, for example, the study by “Cox, D. R. (1992). Causality: some statistical aspects. Journal of the Royal Statistical Society: Series A (Statistics in Society), 155(2), 291-301” and the study by “Wunsch, G., Russo, F., & Mouchart, M. (2010). Do we necessarily need longitudinal data to infer causal relations?. Bulletin of Sociological Methodology/Bulletin de Méthodologie Sociologique, 106(1), 5-18.”

METHODS

In the subsection of “Sample and participant selection” it is important to specify where the clinical and healthy sample has been recruited.

It is also important to better clarify the specific inclusion/exclusion criteria of both the clinical group and the healthy group.

What did you use to collect demographic information? Did you build a questionnaire? Through an interview? It is important to specify this information.

The authors said that they used the Child Feeding Questionnaire for adolescents (CFQ-A), and that, for this study, the questionnaire was translated into German language. There's no German validated version? This is an important limitation of the study, which must be added among the limitations.

For all instruments, it is important to specify if you have used a German validated version. Moreover, the psychometric properties of the original version of the instruments should be reported.

In the section on statistical analysis, it is important to highlight which analyses have been carried out considering the two different groups (ED and HC) and which ones considering the total sample.

RESULTS

The section “Sample description” is confusing and needs to be reorganized. I suggest you initially describe the initial sample you recruited (how many clinical subjects? How many regulatory subjects?). then specify how many cases were excluded from the clinical group, specifying how many for each exclusion criterion. Next, describe how many subjects were excluded from the regulatory group, specifying how many for each exclusion criterion. Finally, describe the characteristics of the final sample, for each of the two groups. Furthermore, it is not clear how many subjects each group is composed of. This information should be added.

In the abstract, the authors referred that the age of the study participants was between 16 and 30 years, but in the results they reported a range between 16 and 26 years. What is the age range of the sample?

Moreover, in the description of mediation results, it is necessary to specify whether the analyses were carried out on the total sample and why.

In the comments of table 1, it is not clear what the phrase "Based on values from 0 = never to 4 = always" refers to. At the same time, in the comments of table 3, it is not clear what the phrase “Based on values from 1 = never / does not apply at all to 10 5 = always / applies entirely” refers to.

DISCUSSION

As suggested for the introduction, it is important that the theoretical approach of the authors is best outlined also in the discussion, and that the focus of the work is well exposed.

Most of the results are discussed on the basis of the results of previous studies. However, overall, there is a lack of hypotheses underlying the findings. For example, one of the main strengths of the study was to have considered the role played by the paternal contribution. However, the authors do not refer to studies in the previous literature (Page 14, line 18). I suggest to cite them. At this regard, see for example, the study by Cerniglia et al. (2017), Family profiles in eating disorders: family functioning and psychopathology. Psychology research and behavior management, 10, 305.

Overall, It is necessary to enrich the discussion of each result of the study by providing possible theoretical explanations.

Page 15, Line 7, there is a typing error. “Stenbæk et al. (8)” should be “Stenbæk et al. (8)”

Reviewer #2: Dear editor,

Thank you very much for the invitation to review the manuscript entitled “Parental feeding practices and the relationship with parents in young women with eating disorders: A case control study”.

I read with very much interest the paper, that is focused on a topic of growing importance for the clinic and research.

Actually, in fact, eating disorders represent a very relevant issue in the international scientific literature. In particular, many researchers have concentrated their interest on the phenomenon of problematic behaviors in various specific developmental ages, such as adolescence and youth.

My overall impression on the manuscript is positive.

Firstly, the authors in their work discuss the topic of eating disorders in adolescents and young women highlighting the relevance of family environment in this complex type of disease. Moreover, specifically, they focus the interest of research on family relationships and parental feeding practices as very relevant aspects that should be taken into account to better understand the son’s sufferance and to organize appropriate intervention strategies.

The writing is overall understandable and the study appears to be sound (form and contents are quite clear). The introduction section, the general aim and results are clearly recognized. Moreover, the use of written English is quite good and clear.

These elements as a whole represents a manuscript’s strengths.

For the above considerations, I think that this work can improve the field of eating disorders, precisely the topic of the relationship between parental dynamics and psychopathological offspring’s well-being.

Nevertheless, I would like to discuss some areas of improvement in the manuscript, so that the authors can use the following comments to review their paper.

Title

The title in the full version is perhaps too long and lengthy. The expression Parental feeding practices and the relationship with parents could be replaced with a more general expression that indicate the core theme of the study (family relationships?-family interaction? ecc..)

Maybe the words young women, in the title, don’t refer to the real sample of the study, that also consists in adolescents?

Abstract

Since the age of subjects participating in the study ranges from 16 to 30 years old, the authors should better specify in the text the use of the terms woman-women and adolescents.

Introduction

In the first part of introduction, some aspects appear to be unclear. Authors should include more information about the developmental age of adolescence and youth, also specifying epidemiological data (for instance, about gender differences).

Thus, in the introduction it may be useful to insert a first part on the topic of adolescence by indicating the crucial aspects of this phase of lifecycle, such as the psychological and emotional functioning, the family relationships, the social modifications ecc…). Moreover, I suggest to better clarify the psychological and relational adolescent’s conditions, indicating some empirical contributions on risks and protective factor in adolescence and on family elements.

Moreover, the authors should stress all these aspects of their work by pointing out the focus on relationships in eating disorders, also deepening the related literature: the interest in relationships allows to focus the eating problematic behavior in female adolescents and young women in a systemic perspective, according to which the individual symptom can be understood only within the relational dynamics among family members.

These topics could be better discussed with these works

- Erriu, M., Cimino, S., & Cerniglia, L. (2020). The Role of Family Relationships in Eating Disorders in Adolescents: A Narrative Review. Behavioral Sciences, 10(4), 71.

- Tafà, M., Cimino, S., Ballarotto, G., Bracaglia, F., Bottone, C., & Cerniglia, L. (2017). Female adolescents with eating disorders, parental psychopathological risk and family functioning. Journal of Child and Family Studies, 26(1), 28-39.

-Treasure, J.; Duarte, T.A.; Schmidt, U. Eating disorders. Lancet 2020

In addition, since binge eating disorder (BED) is a specific diagnostic category defined in the DSM V (Apa 2013), it should be rather suggested to write a brief definition of BED clinical features and then to better specify the sample criteria selection (the reasons for the inclusion of anorexia and bulimia in relation to BED).

Finally, the authors should clarify the theoretical framework adopted in relation to the topic of eating disorder in adolescence and youth. Starting from this point, authors should be better articulate the specific hypothesis and objectives of the research.

Method

The section on method could be improved in the choice of titles to be given to subsections.

A possible articulation could be the following:

Research Methods

-Subjects and procedure

-Measures

-Statistical analysis

It would be convenient to better define the selection of the specific sample study (why the age of adolescents starts at 16? And why the age range is from 16 to 30 years?).

Moreover, I suggest to add some additional information about the recruitment procedure.

Results and conclusions

I think the link between the introduction section and the final section is not very clear.

The introduction section should contain a more precise definition of the hypotheses of the research, objectives and variables. The conclusion section should discuss results in relation to the premises. More precisely, in the conclusions the authors should better explain how the findings are related to their initial assumptions.

Finally, since the study is not defined as longitudinal but retrospective research, the authors should better discuss this element.

6. PLOS authors have the option to publish the peer review history of their article (what does this mean?). If published, this will include your full peer review and any attached files.

Reviewer #1: No

Reviewer #2: No

---

## [Author Response · Author response to Decision Letter 0]

27 Oct 2020

When submitting your revision, we need you to address these additional

requirements.

1. Please ensure that your manuscript meets PLOS ONE's style

requirements, including those for file naming.

Dear editorial staff at PLOSone! Thank you for bringing this to our attention: We made changes to meet PLOS ONE`s style (e.g. adaptation of level headings).

2. Please state in your methods section whether you obtained consent from parents or guardians of the minors (those aged <18) included in the study or whether the research ethics committee or IRB approved the lack of parent or guardian consent.

Dear editorial team at PLOSone: Thank you! We have now tried to more clearly state the information required. Please see page 7, section „Study Setting“

3. We note that you have indicated that data from this study are available upon request. PLOS only allows data to be available upon request if there are legal or ethical restrictions on sharing data publicly. For information on unacceptable data access restrictions, please see

Dear editorial team at PLOSone: Thank you for this additional information! We have now tried to include a more detailed information on data availability (see page 23, Section „Availability of Data and Materials“, lines 13-16)

b) If there are no restrictions, please upload the minimal anonymized data set necessary to replicate your study findings as either Supporting Information files or to a stable, public repository and provide us with the relevant URLs, DOIs, or accession numbers. Please see http://www.bmj.com/content/340/bmj.c181.long for guidelines on how to de-identify and prepare clinical data for publication. For a list of acceptable repositories, please see http://journals.plos.org/plosone/s/data-availability#loc-recommended-repositories [3].

Dear editorial team at PLOSone: Thank you! We have updated the author information accordingly!

5. Your ethics statement must appear in the Methods section of your manuscript. If your ethics statement is written in any section besides the Methods, please move it to the Methods section and delete it from any other section. Please also ensure that your ethics statement is included in your manuscript, as the ethics section of your online submission will not be published alongside your manuscript.

Dear editorial team at PLOSone: Thank you! We have now tried to make the information more visible. (see page 7, lines 9-16)

 

Reviewers' comments:

Reviewer #1:

Thank you for the opportunity of revising this manuscript titled “Parental feeding practices and the relationship with parents in young women with eating disorders: A case control study’. The study took in consideration, in a sample of young women with and without an eating disorder, the possible association between eating disorders and characteristics of the relationship with parents and the parental feeding practices in early adolescence.

Dear Reviewer: We thank you for taking the time to send a very thoughtful and thorough review! We have strived to enact changes according to your comments. In their entirety, we feel, that the paper has gained in preciseness and quality by doing so! For all comments, that were answered positively and in full, we have marked these in *green* color.

I read the article and I think the focus is very interesting.

Thank you!

Despite this, several structural revisions are necessary. So, I think that it can be published in this Journal, but with major revision. In particular I think it is important to give evidence to the theoretical model, also illustrating a major study of recent literature. Please find below some comments.

We again, thank you for a very thoughtful and thorough review!

INTRODUCTION

In the introduction, the concept of EDS is introduced, but it is not adequately explained, referring also to the most recent diagnostic classifications (e.g. DSM-5).

Thank you! It is correct, that we had not previously described the individual entities of eating disorders (in part so “safe space”). We have now included – at the beginning of the Introduction – paragraphs concerning the eating disorder definitions

Moreover, the adolescent population is the object of interest. However, the authors reported prevalence rates without referring to this specific developmental stage. It's important to report the prevalence rates and the clinical relevance of the phenomenon in adolescence. Moreover, I suggest to cite previous studies that have shown that the peaks of EDS onset are in the adolescent phase. See, for example, the study by Poppe I, Simons A, Glazemakers I, Van West D. Early-onset eating disorders:a review of the literature. Tijdschr Psychiatr. 2015;57(11):805–814.

Thank you! We have now strived to present a more nuanced presentation of prevalence according to the age group of interest and have now also included references to risinig rates of prevalence.

In addition, in the introduction is not specified why adolescence represents a phase of risk for the onset and maintenance of Eating disorders. It would be important to underline the peculiarities of this specific evolutionary stage. We recommend, for example: Baker et al.,

(2012), “Pubertal development predicts eating behaviors in adolescence” Or Ballarotto et al.(2017). Does alexithymia have a mediating effect between impulsivity and emotional-behavioural functioning in adolescents with binge eating disorder?. Clinical Neuropsychiatry, 14(4).

Thank you! You are correct in stating the importance of these aspects! We have created a section on adolescents as a risk factor and the mentioned author’s papers in the introduction as well as the discussion!

Moreover, from the outset, the theoretical framework should be clear. The authors should describe from the beginning the theoretical framework from which they start for their own study.

We have now included a section on theoretical framework.

On the other hand, the links between parents and children are highlighted, but the influence and emotional exchange between parents and children, linked to eating patterns, is not clear. It can be helpful for a review, see the study: Cimino et al.(2018). Impact of parental binge eating disorder: exploring children's emotional/behavioral problems and the quality of parent–child feeding interactions. Infant mental health journal, 39(5), 552-568.

Thank you for this suggestion! We have included a section on this and we feel, that it has gained in relevance by doing so!

The aims of the study are not clear. For example, the statistical analysis section showed that you also tested a mediation model, but it is not clear on the basis of which hypotheses. I suggest that you reorganize the last part of the introduction, describing the objectives of the study in detail. For each aim, it is also important to specify what the hypotheses are, and based on which previous studies and theoretical perspective.

Thank you for this suggestion! We have seen, that – in some areas – a further clarification was necessary. We have now reorganized the final passage of the introduction. Furthermore, the aims section has been specified and clarified on which previous studies they are based. In its entirety, we hope, that our revised manuscript more closely reflects our intentions.

This is not a longitudinal study, so authors should clarify already in the introduction because, and on the basis of which literature, it is possible to draw cause-effect conclusions in retrospective or cross-sectional studies. See, for example, the study by “Cox, D. R. (1992). Causality: some statistical aspects. Journal of the Royal Statistical Society: Series A (Statistics in Society), 155(2), 291-301” and the study by “Wunsch, G., Russo, F., & Mouchart, M. (2010). Do we necessarily need longitudinal data to infer causal relations?. Bulletin of Sociological Methodology/Bulletin de Méthodologie Sociologique, 106(1), 5-18.”

Thank you for this suggestion! It has been clarified that it is possible to draw cause-effect conclusions not only in longitudinal studies, by using a structural modelling strategy.

METHODS

In the subsection of “Sample and participant selection” it is important to specify where the clinical and healthy sample has been recruited. It is also important to better clarify the specific inclusion/exclusion criteria of both the clinical group and the healthy group.

Thank you for pointing out, that this information had been presented in a too short and incomplete form in the previous version! We have now expanded this section, describing inclusion/exclusion and recruitment!

What did you use to collect demographic information? Did you build a questionnaire? Through an interview? It is important to specify this information.

Again, we need to thank you to point out that this information had been omitted previously. We have now added a paragraph in the Methods-section describing the acquisition of demographic data. We have used our own additional one-page questionnaire for this purpose.

The authors said that they used the Child Feeding Questionnaire for adolescents (CFQ-A), and that, for this study, the questionnaire was translated into German language. There's no German validated version? This is an important limitation of the study, which must be added among the limitations. For all instruments, it is important to specify if you have used a German validated version. Moreover, the psychometric properties of the original version of the instruments should be reported.

Pointing out the possible limitation is of course correct! We had previously not mentioned the limitation as extensive as in the current version: the CFQ-A was used in a German translation made only for this study. In the interest of future studies, we have included the psychometric properties of the original as well as the translated version. We have, as suggested, included a paragraph in the limitations-section of the paper!

For all other tests, the psychometric properties were already included in the paper.

In the section on statistical analysis, it is important to highlight which analyses have been carried out considering the two different groups (ED and HC) and which ones considering the total sample.

Thank you! That is a very observant and valid point! We have now included statement on how many participants were included in the individual analysis

RESULTS

The section “Sample description” is confusing and needs to be reorganized. I suggest you initially describe the initial sample you recruited (how many clinical subjects? How many regulatory subjects?). then specify how many cases were excluded from the clinical group, specifying how many for each exclusion criterion. Next, describe how many subjects were excluded from the regulatory group, specifying how many for each exclusion criterion. Finally, describe the characteristics of the final sample, for each of the two groups. Furthermore, it is not clear how many subjects each group is composed of. This information should be added.

Thank you! We like to extend our gratitude to you for taking the time to help us in organizing the text in a more appropriate fashion! We have restructured the paragraph as mentioned!

In the abstract, the authors referred that the age of the study participants was between 16 and 30 years, but in the results they reported a range between 16 and 26 years. What is the age range of the sample?

That is very observant! In an earlier version of the text it had – most likely – been rounded up! We have now corrected this mistake in the abstract!

Moreover, in the description of mediation results, it is necessary to specify whether the analyses were carried out on the total sample and why.

Thank you! That is correct! As stated in a previous comment-answer, we have now included the requested information.

In the comments of table 1, it is not clear what the phrase "Based on values from 0 = never to 4 = always" refers to. At the same time, in the comments of table 3, it is not clear what the phrase “Based on values from 1 = never / does not apply at all to 10 5 = always / applies

entirely” refers to.

Again, that is very observant of you! Thank you for pointing this out! We have now re-phrased and reformatted the section concerned.

 

DISCUSSION

As suggested for the introduction, it is important that the theoretical approach of the authors is best outlined also in the discussion, and that the focus of the work is well exposed.

Thank you!

Most of the results are discussed on the basis of the results of previous studies. However, overall, there is a lack of hypotheses underlying the findings. For example, one of the main strengths of the study was to have considered the role played by the paternal contribution. However, the authors do not refer to studies in the previous literature (Page 14, line 18). I suggest to cite them. At this regard, see for example, the study by Cerniglia et al. (2017), Family

profiles in eating disorders: family functioning and psychopathology. Psychology research and behavior management, 10, 305.

Thank you for this suggestion! We have now included the studies mentioned in our discussion and expanded the concerning sections!

Overall, It is necessary to enrich the discussion of each result of the study by providing possible theoretical explanations.

Thank you! We have now expanded the theoretical background!

Page 15, Line 7, there is a typing error. “Stenbæk et al. (8)”

should be “Stenbæk et al. (8)”

Typing was checked again for all authors and their citations.

 

Reviewer #2: Dear editor,

Thank you very much for the invitation to review the manuscript entitled “Parental feeding practices and the relationship with parents in young women with eating disorders: A case control study”.

I read with very much interest the paper, that is focused on a topic of growing importance for the clinic and research. Actually, in fact, eating disorders represent a very relevant issue in the international scientific literature. In particular, many researchers have concentrated their interest on the phenomenon of problematic behaviors in various specific developmental ages, such as adolescence and youth.

Thank you! We are as well very infested in this field and feel that it is sometimes not as much in the clinical focus as it should be!

My overall impression on the manuscript is positive.

Firstly, the authors in their work discuss the topic of eating disorders in adolescents and young women highlighting the relevance of family environment in this complex type of disease. Moreover, specifically, they focus the interest of research on family relationships and parental feeding practices as very relevant aspects that should be taken into account to better understand the son’s sufferance and to organize appropriate intervention strategies.

Thank you!

The writing is overall understandable and the study appears to be sound (form and contents are quite clear). The introduction section, the general aim and results are clearly recognized. Moreover, the use of written English is quite good and clear.

These elements as a whole represents a manuscript’s strengths. For the above considerations, I think that this work can improve the field of eating disorders, precisely the topic of the relationship between parental dynamics and psychopathological offspring’s well-being.

Again, thank you! We very much appreciate these kind words of support!

Nevertheless, I would like to discuss some areas of improvement in the manuscript, so that the authors can use the following comments to review their paper.

Dear Reviewer! We have strived to follow all of your suggestions and we thank you for your very focused – but still comprehensive! – suggestions! We feel, that the overall quality of the paper was greatly improved!

Title

The title in the full version is perhaps too long and lengthy. The expression Parental feeding practices and the relationship with parents could be replaced with a more general expression that indicate the core theme of the study (family relationships?-family interaction? ecc..) Maybe the words young women, in the title, don’t refer to the real sample of the study, that also consists in adolescents?

That is a very observant and important point. Indeed, we have now, throughout the paper, used more precise wording concerning the study sample and have, thus, also changed the title to incorporate your suggestion!

Abstract

Since the age of subjects participating in the study ranges from 16 to 30 years old, the authors should better specify in the text the use of the terms woman-women and adolescents.

Thank you! As previously mentioned, we have followed your suggestion and changed the wording wherever applicable!

Introduction

In the first part of introduction, some aspects appear to be unclear. Authors should include more information about the developmental age of adolescence and youth, also specifying epidemiological data (for instance, about gender differences). Thus, in the introduction it may be useful to insert a first part on the topic of adolescence by indicating the crucial aspects of this phase of lifecycle, such as the psychological and emotional functioning, the family relationships, the social modifications ecc…). Moreover, I suggest to better clarify the psychological and relational adolescent’s conditions, indicating some empirical contributions on risks and protective factor in adolescence and on family elements. Moreover, the authors should stress all these aspects of their work by pointing out the focus on relationships in eating disorders, also deepening the related literature: the interest in relationships allows to focus the eating problematic behavior in female adolescents and young women in a systemic perspective, according to which the individual symptom can be understood only within the relational dynamics among family members.

These topics could be better discussed with these works

- Erriu, M., Cimino, S., & Cerniglia, L. (2020). The Role of Family Relationships in Eating Disorders in Adolescents: A Narrative Review. Behavioral Sciences, 10(4), 71.

Tafà, M., Cimino, S., Ballarotto, G., Bracaglia, F., Bottone, C., & Cerniglia, L. (2017). Female adolescents with eating disorders, parental psychopathological risk and family functioning. Journal of Child and Family Studies, 26(1), 28-39.

Treasure, J.; Duarte, T.A.; Schmidt, U. Eating disorders. Lancet 2020

Thank you for this thorough review and discussion of possible points of imprevements! We have tried to follow your suggestions and adapted the text now – in some areas extensively. And added the citations suggested!

In addition, since binge eating disorder (BED) is a specific diagnostic category defined in the DSM V (Apa 2013), it should be rather suggested to write a brief definition of BED clinical features and then to better specify the sample criteria selection (the reasons for the inclusion of

anorexia and bulimia in relation to BED).

Thank you for the suggestion! As we have only included AN and BN, but no patients with BED, we have now included a section on criteria of BED and the lack of its inclusion as a possible limitation!

Finally, the authors should clarify the theoretical framework adopted in relation to the topic of eating disorder in adolescence and youth. Starting from this point, authors should be better articulate the specific hypothesis and objectives of the research.

Thank you! As also the other reviewer has pointed out a similar limitation, we have taken the time to rethink how our work is perceived and how we should adapt the presentation of it accordingly to better represent our indented meaning! We have, thus, undertaken a rather extensive revision and feel that, following the suggestions in the reviewprocess, the readability and quality of the paper was improved.

Method

The section on method could be improved in the choice of titles to be given to subsections.

A possible articulation could be the following:

Research Methods

-Subjects and procedure

-Measures

-Statistical analysis

Thank you! We have changed the formatting accordingly!

It would be convenient to better define the selection of the specific sample study (why the age of adolescents starts at 16? And why the age range is from 16 to 30 years?). Moreover, I suggest to add some additional information about the recruitment procedure.

That is a very valid point! We have now added a more extensive description of the process including the inclusion as well as exclusion criteria and how recruitment was undertaken.

We have, also, more clearly defined the age groups. The reason for the age range used was due to pragmatic reasons as these are the ranges most often present in our wards (most often present and of legal age of consent in Austria).

Results and conclusions

I think the link between the introduction section and the final section is not very clear. The introduction section should contain a more precise definition of the hypotheses of the research, objectives and variables. The conclusion section should discuss results in relation to the premises. More precisely, in the conclusions the authors should better explain how the findings are related to their initial assumptions. Finally, since the study is not defined as longitudinal but retrospective research, the authors should better discuss this element.

Thank you for these two suggestions! Indeed, we have seen that our presentation was not optimal, and we have now extensively rewritten some parts according to your suggestions and hope that you will also find, as do we, that the study presentation has been greatly improved by your suggestions!

---

## [Editor Report · Decision Letter 1]

4 Nov 2020

Parental feeding practices and the relationship with parents in female adolescents and young adults with eating disorders: A case control study

PONE-D-20-11413R1

Dear Authors,

We’re pleased to inform you that your manuscript has been judged scientifically suitable for publication and will be formally accepted for publication once it meets all outstanding technical requirements.

Kind regards,

Silvia Cimino

Academic Editor

PLOS ONE
---

## [Editor Report · Acceptance letter]

10 Nov 2020

PONE-D-20-11413R1 

Parental feeding practices and the relationship with parents in female adolescents and young adults with eating disorders: A case control study 

Dear Dr. Werneck:

I'm pleased to inform you that your manuscript has been deemed suitable for publication in PLOS ONE. Congratulations! Your manuscript is now with our production department. 

Kind regards, 

on behalf of

Professor Silvia Cimino 

Academic Editor

PLOS ONE